# Trends in social determinants of child health and perinatal outcomes in European countries 2005–2015 by level of austerity imposed by governments: a repeat cross-sectional analysis of routinely available data

Luis Rajmil,[1] David Taylor-Robinson,[2] Geir Gunnlaugsson,[3] Anders Hjern,[4] Nick Spencer[5]

For numbered affiliations see end of article.

**Correspondence to**
Dr Luis Rajmil;
12455lrr@comb.cat

## ABSTRACT

**Objective** To assess whether the level of austerity implemented by national governments was associated with adverse trends in perinatal outcomes and the social determinants of children's health (SDCH) in rich countries

**Design** Longitudinal ecological study of country-level time trends in perinatal outcomes and SDCH and from 2005 to 2015.

**Setting and participants** 16 European countries using available data from the International Monetary Fund, the Organisation for Economic Co-operation and Development and Eurostat.

**Main outcome measures** Trends in perinatal outcomes (low birth weight (LBW); infant mortality) and the SDCH: child poverty rates; severe material deprivation in families with primary education; preschool investment in three time periods: 2005–2007, 2008–2010 and 2012–2015. Outcomes were compared according to the cyclically adjusted primary balance (CAPB, differences between 2013 and 2009) as a measure of austerity, stratified in tertiles. Generalised estimating equation models of repeated measures were used to assess time trend differences in three periods.

**Results** Countries with higher levels of austerity had worse outcomes, mainly at the last study period. Material deprivation increased during the period 2012–2015 in those countries with higher CAPB (interaction CAPB-period 2012–2015, B: 5.62: p<0.001), as did LBW (interaction CAPB-period 2012–2015, B: 0.25; p=0.004).

**Conclusions** Countries that implemented more severe austerity measures have experienced increasing LBW, and for families with primary education also increasing material deprivation, worsening the negative impact of economic crisis. Reversing austerity policies that impact children is likely to improve child health outcomes.

## INTRODUCTION

The global financial crisis starting in 2008 has had a great economic impact around the world, although the impact in each country

### Strengths and limitations of this study

► This is one of the first studies to analyse trends in social determinants of children's health (SDCH) and perinatal indicators starting before the economic crisis.
► The main strengths of the study include the use of nationally representative data with standardised and comparable definitions and methods, and a study design that allowed us to analyse time trends starting in 2005, 3 years before the economic crisis, through to 2015.
► Countries included in the study may not allow the results to be generalised.
► Variables used to assess austerity, SDCH and perinatal outcomes may have limited ability to discriminate specific measures taken by governments.

depends on several factors, such as the starting point, mechanisms of social protection and social transfers, and the measures adopted by governments to deal with the crisis.[1] During the period 2008–2010, the initial financial crisis was followed by a downturn in economic activity, resulting in evictions, foreclosures and prolonged unemployment in many countries. From 2010 to 2011 onwards, the pressure to adopt and enforce austerity measures, mainly within the European Union (EU) and worldwide, has been levied by global financial institutions such as the International Monetary Fund (IMF).[2–4] These austerity measures taken by governments were neither homogeneous nor similarly implemented in rich countries; some protected public sector programmes and systems while others instituted large budget cuts in education, health and other public services.

Assessing the health effects of the crisis is challenging given the difficulty establishing association of exposure and outcomes. There is a lag time between exposure to the effects of a severe economic downturn and adverse health outcomes[5] and, in the case of exposure in early childhood, the adverse effects may not emerge until adulthood.[6] Childhood is an especially vulnerable period to the main determinants of health, such as living conditions, family income, employment, education, housing, access to health services, among other social determinants of child health (SDCH) to individual life style factors. All factors related to financial capital (ie, family income), human capital (ie, education) and social capital (family working conditions) have potential influence on future child health and development.[7 8] Moreover, there is overwhelming evidence for the profound effects of social factors and SDCH on health throughout childhood and into adulthood.[9–11]

Findings from studies to date on the impact of the economic crisis on SDCH and perinatal outcomes, carried out at national and international level show fetal and childhood populations as population groups most affected by the crisis.[12–16] Vulnerability starts in the prenatal period with adverse effects on perinatal outcomes.[17–19] Most studies of the impact of economic crisis on health have not distinguished between economic crises themselves and policy responses to these crises.[6] Furthermore, there is a paucity of literature on the impact of austerity measures imposed by government in response to the economic crisis on perinatal outcomes and SDCH.[20] The analysis of time trends beginning before the crisis until the postcrisis period allows evaluation of political responses and their impact on SDCH and perinatal health.

The aim of the study was therefore to assess time trends in perinatal outcomes and SDCH by level of austerity enforced by governments. We hypothesise that those countries that implemented and maintained high levels of austerity would show statistically significant differences in adverse trends in perinatal outcomes and SDCH compared with countries imposing lower levels of austerity.

## METHODS
We undertook a longitudinal ecological study of trends in perinatal outcomes and SDCH at country level, and assessed how these trends varied according to the level of austerity measures implemented. Sixteen European countries from the European Economic Area were included in the study, on the basis of routinely available data for the period 2005–2015 pertaining to perinatal outcomes and SDCH. Luxembourg was excluded due to its high level of economic development, which not necessarily reflects the real wealth of residents. Postcommunist countries were also excluded. The included countries are evenly distributed across low, intermediate and high austerity measures.

## Outcome variables
Perinatal outcomes: Data on perinatal outcomes were taken from the Organisation for Economic Co-operation and Development (OECD) Family Database (http://www.oecd.org/els/family/database.htm).
1. Low birth weight (LBW) was defined as the number of live births weighing less than 2500 g divided by the total number of live births.
2. Infant mortality (IM): deaths of children aged less than 1 year per 1000 live births (no minimum threshold of gestation period or birth weight).

## Social determinants of child health
1. Child poverty was defined as the percentage of children living in households with income below 60% of the median. Data were taken from the EU-Survey of Income and Living Conditions (EU-SILC) (http://ec.europa.eu/eurostat/web/microdata/european-union-statistics-on-income-and-living-conditions).
2. Severe material deprivation rate was defined as the proportion of children under 18 years of age living in families with primary level of education which cannot afford to pay for at least four out of the nine items considered basic, such as having arrears on mortgage or rent payments, utility bills, hire purchase instalments or other loan payments; not being able to afford 1-week annual holiday away from home; not being able to afford a meal with meat, chicken, fish (or vegetarian equivalent) every second day, etc. Data were taken from the EU-SILC database.

**Table 1** Scores of the cyclically adjusted primary balance (CAPB) and stratified according to the level of austerity

| Country | CAPB | Level of austerity |
| --- | --- | --- |
| Denmark | 0.02 | Low |
| Finland | −0.1 | Low |
| Germany | 0.08 | Low |
| Norway | 0.14 | Low |
| Sweden | −0.43 | Low |
| Austria | 0.58 | Intermediate |
| Belgium | 0.35 | Intermediate |
| France | 0.52 | Intermediate |
| Italy | 0.64 | Intermediate |
| Netherlands | 0.74 | Intermediate |
| Greece | 3.43 | High |
| Iceland | 1.75 | High |
| Ireland | 1.83 | High |
| Portugal | 1.68 | High |
| Spain | 1.93 | High |
| UK | 0.86 | High |

The average annual change in the difference between taxes and non-interest spending 2013–2009 would be if the economy were at full employment. Higher score corresponds to higher level of austerity.

**Table 2**  Low birth weight (LBW) and infant mortality (IM) in three time periods (2005–2007/2008–2010/2012–2015), by country and according to the level of austerity in tertiles (low, intermediate and high)

| | Perinatal outcomes | | | | | |
| --- | --- | --- | --- | --- | --- | --- |
| | LBW (%) | | | IM (‰) | | |
| Level of austerity | 2005–2007 | 2008–2010 | 2012–2015 | 2005–2007 | 2008–2010 | 2012–2015 |
| Low | | | | | | |
| Denmark | 5.30 | 5.30 | 5.25 | 3.96 | 3.50 | 3.65 |
| Finland | 4.23 | 4.23 | 4.15 | 2.83 | 2.50 | 2.02 |
| Germany | 6.83 | 6.86 | 6.75 | 3.90 | 3.46 | 3.27 |
| Norway | 4.98 | 5.20 | 4.62 | 3.13 | 2.86 | 2.40 |
| Sweden | 4.20 | 4.26 | 4.35 | 2.56 | 2.50 | 2.50 |
| Total | 5.10 | 5.17 | 5.02 | 3.28 | 2.96 | 2.76 |
| Intermediate | | | | | | |
| Austria | 7.03 | 7.06 | 6.67 | 3.83 | 3.80 | 3.10 |
| Belgium | 6.96 | 6.93 | 6.90 | 3.90 | 3.63 | 3.50 |
| France | 6.65 | 6.73 | 6.33 | 3.80 | 3.76 | 3.57 |
| Italy | 6.76 | 7.10 | 7.35 | 3.20 | 3.10 | 2.87 |
| Netherlands | 6.76 | 5.83 | 5.80 | 4.46 | 3.80 | 3.80 |
| Total | 6.83 | 6.73 | 6.81 | 3.84 | 3.61 | 3.32 |
| High | | | | | | |
| Greece | 8.86 | 9.33 | 9.32 | 3.66 | 3.20 | 3.60 |
| Iceland | 3.90 | 3.83 | 4.10 | 1.90 | 2.16 | 1.80 |
| Ireland | 4.93 | 5.00 | 5.55 | 3.63 | 3.43 | 3.42 |
| Portugal | 7.63 | 8.06 | 8.70 | 3.40 | 3.13 | 3.02 |
| Spain | 7.33 | 7.70 | 7.80 | 3.53 | 3.23 | 2.85 |
| UK | 7.36 | 7.03 | 6.95 | 4.90 | 4.43 | 3.92 |
| Total | 6.66 | 6.82 | 7.07 | 3.5 | 3.26 | 3.1 |

3. Preschool investment (0–5 years old) was calculated as the annual percentage of gross domestic product (GDP) for each country. Data were taken from the OECD database.

### Independent variable of interest
#### Austerity assessment
An indicator based on the cyclically adjusted primary balance (CAPB) published by the IMF[21] was used to analyse and identify the extent of austerity policy responses to the crisis in each participating country during the period 2009–2013[22], in general the last year with major spending cuts. The CAPB represents the cyclical component of the overall fiscal balance, computed as the difference between cyclical revenues and cyclical expenditures. It represents the average annual change in the CAPB, an estimate of what the difference between taxes and non-interest spending would be if the economy were at full employment. A high score equates to a higher level of austerity. CAPB was divided into tertiles representing high, medium and low level of austerity.

#### Covariable
Time period (2005 to the latest available data) was stratified in three time periods: 2005–2007 (precrisis); 2008–2010 (economic crisis); 2012–2015 (austerity and welfare period or postcrisis period).

### Data analysis
The analysis progressed in three stages. First, we undertook a descriptive analysis of longitudinal trends in perinatal outcomes and the SDCH.

Second, we assessed the relationship between austerity category, and change in perinatal outcomes and SDCH, stratified in the three time periods.

Finally, we used a longitudinal generalised estimating equations (GEE) model, based on robust SEs,[23] to allow for analysis of our correlated repeated outcome measurements (SDCH and perinatal outcomes; see online supplementary figure 1). GEE has been advocated as a tool for evaluating policy change and natural experiments.[24] The independent variables included in the models were time period and CAPB. Time period was stratified in three time periods for our final model. Interaction terms were also included in the model to assess the influence of

**Table 3** SDCH in three time periods (2005–2007/2008–2010/2012–2015), by country and according to the level of austerity in tertiles (low, intermediate and high)

| Level of austerity | SDCH | | | | | | | | |
|---|---|---|---|---|---|---|---|---|---|
| | Child poverty (%) | | | Material deprivation (%) | | | Preschool investment (% of GDP) | | |
| | 2005–2007 | 2008–2010 | 2012–2015 | 2005–2007 | 2008–2010 | 2012–2015 | 2005–2007 | 2008–2010 | 2012–2015 |
| **Low** | | | | | | | | | |
| Denmark | 8.33 | 8.81 | 8.30 | 14.20 | 7.53 | 13.80 | 1.29 | 1.26 | 1.37 |
| Finland | 8.96 | 10.50 | 8.87 | 12.80 | 14.06 | 11.87 | 0.88 | 0.99 | 1.10 |
| Germany | 11.50 | 13.56 | 13.40 | 24.83 | 30.00 | 30.52 | 0.37 | 0.42 | 0.55 |
| Norway | 8.63 | 8.90 | 8.37 | 8.56 | 11.50 | 14.00 | 0.83 | 1.12 | 1.22 |
| Sweden | 10.36 | 11.23 | 13.5 | 11.73 | 10.20 | 10.00 | 1.28 | 1.47 | 1.60 |
| Total | 9.55 | 10.59 | 10.48 | 14.42 | 14.65 | 16.03 | 0.93 | 1.05 | 1.16 |
| **Intermediate** | | | | | | | | | |
| Austria | 12.63 | 14.90 | 14.95 | 16.49 | 18.63 | 21.35 | 0.28 | 0.38 | 0.48 |
| Belgium | 14.60 | 14.73 | 15.72 | 22.36 | 21.0 | 24.35 | 0.60 | 0.64 | 0.75 |
| France | 13.40 | 14.83 | 16.37 | 16.53 | 22.53 | 23.70 | 1.10 | 1.14 | 1.25 |
| Italy | 22.63 | 22.13 | 23.77 | 15.70 | 17.86 | 29.12 | 0.52 | 0.54 | 0.53 |
| Netherlands | 11.70 | 11.32 | 11.40 | 10.30 | 9.06 | 9.27 | 0.54 | 0.82 | 0.71 |
| Total | 14.9 | 15.5 | 16.4 | 16.23 | 17.81 | 21.55 | 0.60 | 0.70 | 0.74 |
| **High** | | | | | | | | | |
| Greece | 21.96 | 22.43 | 27.02 | 20.83 | 27.33 | 50.82 | – | – | – |
| Iceland | 9.50 | 9.13 | 8.80 | 7.76 | 2.60 | 7.70 | 1.30 | 1.57 | 1.81 |
| Ireland | 18.33 | 16.26 | 17.12 | 17.53 | 19.10 | 26.12 | 0.28 | 0.43 | 0.50 |
| Portugal | 18.60 | 19.80 | 21.97 | 12.93 | 14.33 | 18.60 | 0.35 | 0.37 | 0.37 |
| Spain | 22.60 | 24.46 | 26.67 | 9.23 | 11.16 | 17.52 | 0.43 | 0.52 | 0.53 |
| UK | 19.96 | 18.93 | 18.10 | 22.4 | 20.8 | 28.82 | 0.79 | 0.77 | 0.77 |
| Total | 18.4 | 18.5 | 19.9 | 15.09 | 15.80 | 24.93 | 0.63 | 0.73 | 0.79 |

Material deprivation in families with primary education level.

GDP, gross domestic product; SDCH, social determinants of child health.

the time period and the level of austerity. The analysis was conducted using STATA V.12.0.

All procedures were carried out following the data protection requirements of the European Parliament (Directive 95/46/EC of the European Parliament and of the Council of 24 October 1995 on the protection of individuals with regard to the processing of personal data and on the free movement of such data).

### Patient and public involvement

There was no patient or public involvement in the design of this study.

### RESULTS

Higher rates of LBW were seen in Greece, Spain and Portugal, while Iceland showed the lowest rates during the period 2005–2015. Child poverty increased more for Greece and Spain than for the rest of countries, the latter especially showing an increasing trend at the end of the period. By contrast, Finland had lower rates of child poverty with a diminishing trend during the last years. Greece showed an increasing percentage of material deprivation in children from families with primary education level from the year 2009 onwards, while Sweden showed the opposite trend (online supplementary figures 2–6).

Table 1 shows the CAPB 2013–2009 for countries included in the study, and stratified by level of austerity in tertiles. CAPB scores ranged from −0.43 (Sweden) to 3.43 (Greece).

### Perinatal and SDCH outcomes according to the level of austerity

Tables 2 and 3 show perinatal outcomes and SDCH indicators by country, and stratified according to the level of austerity, in three time periods. The high-austerity group presents a trend to increasing LBW rates across the study periods, while the other groups show a slight decrease in 2012–2015. IM shows a continuously diminishing trend in all three groups over the study period. Low-austerity group

**Table 4**  Generalised estimating equation of panel data of perinatal outcomes (LBW and IM) 2005–2015

| | Model 1 (crude) | | Model 2 (adjusted by CAPB) | | Model 3 interactions (time period*CAPB) | |
|---|---|---|---|---|---|---|
| | Coef. | P values | Coef. | P values | Coef. | P values |
| **LBW** | | | | | | |
| Independent variables | | | | | | |
| Time period | | | | | | |
| 2008–2010 | 0.04 | NS | 0.04 | NS | −0.05 | NS |
| 2012–2015 | 0.05 | NS | 0.05 | NS | −0.16 | NS |
| CAPB | | | 0.8 | 0.009 | 0.7 | 0.02 |
| Interaction terms | | | | | | |
| CAPB*2008–2010 | | | | | 0.11 | NS |
| CAPB*2012–2015 | | | | | 0.25 | 0.004 |
| Constant | 6.23 | <0.001 | 5.51 | <0.001 | 5.27 | <0.001 |
| **IM** | | | | | | |
| Independent variables | | | | | | |
| Time period | | | | | | |
| 2008–2010 | −0.256 | <0.001 | −0.25 | <0.001 | −0.25 | 0.002 |
| 2012–2015 | −0.46 | <0.001 | −0.32 | <0.001 | −0.55 | <0.001 |
| CAPB | | | −0.03 | NS | −0.0028 | NS |
| Interaction terms | | | | | | |
| CAPB*2008–2010 | | | | | −0.002 | NS |
| CAPB*2012–2015 | | | | | 0.10 | NS |
| Constant | 3.53 | <0.001 | 3.50 | <0.001 | 3.53 | <0.001 |

Reference category: years 2005–2007.

CAPB, cyclically adjusted primary balance; IM, infant mortality; LBW, low birth weight.

of countries showed lower rates of child poverty and material deprivation than the intermediate and high-austerity group and the rates remained relatively stable across the three study periods (ie, percentage of material deprivation in the low-austerity group was 14.42%; 14.65% and 16.03%; while in the high-austerity group was 15.09%; 15.80% and 24.93%, respectively). Preschool investment increased in the low-austerity group from 0.93% of the GDP to 1.16% in the last period (2012–2015), while in the intermediate and high-austerity groups the increase was lower. Although Iceland is classified by the CAPB as a high-austerity country, it is an outlier as its perinatal and SDCH rates are similar to other Nordic countries in the low-austerity group.

### Multivariate models

Tables 4 and 5 show GEE models for perinatal and SDCH outcomes analysed in three time periods. CAPB was associated with LBW and there was a significant interaction in the period 2012–2015, indicating that LBW increased more in the high-austerity countries (interaction CAPB-period 2012–2015, B: 0.25; p=0.004). No relationship was found between CAPB and IM. Material deprivation increased during the period 2012–2015, and mainly in those countries with higher CAPB (interaction CAPB-period 2012–2015, B: 5.62: P<0.001). No

association was found between preschool investment and CAPB.

### DISCUSSION

To our knowledge, this is the first study exploring the impact of austerity imposed by governments in high-income countries on perinatal outcomes and SDCH above and beyond the impact of the economic crisis itself. The study provides some evidence to suggest that austerity measures themselves have had an impact on children's health, independent of the economic crisis that led to austerity. The results support the hypothesis that those countries that have applied and maintained higher levels of austerity have experienced adverse consequences for children that will likely have implications for the future adult health of a generation.

The results of the present study corroborate other studies at the European general population level.[25] According to the latter, although there are many differences between countries, the analysis suggests that the interaction of fiscal austerity with economic shocks and weak social protection ultimately seems to produce greater social crisis. In general, the initial response to the crisis was increasing public spending during the

**Table 5** Generalised estimating equation of panel data

| | Model 1 (crude) | | Model 2 (adjusted by CAPB) | | Model 3 interactions (time period*CAPB) | |
|---|---|---|---|---|---|---|
| | Coef. | P values | Coef. | P values | Coef. | P values |
| **Child poverty** | | | | | | |
| Time period | | | | | | |
| 2008–2010 | 0.51 | NS | 0.51 | NS | 0.75 | NS |
| 2012–2015 | 1.29 | 0.01 | 1.29 | <0.001 | 0.69 | NS |
| CAPB | | | 3.60 | <0.001 | 3.46 | <0.001 |
| Interaction terms | | | | | | |
| CAPB*2008–2010 | | | | | –0.27 | NS |
| CAPB*2012–2015 | | | | | 0.69 | 0.06 |
| Constant | 14.6 | <0.001 | 11.44 | <0.001 | 11.57 | <0.001 |
| **Material deprivation** | | | | | | |
| Independent variables | | | | | | |
| Time period | | | | | | |
| 2008–2010 | 0.86 | NS | 0.86 | NS | 0.01 | NS |
| 2012–2015 | 5.86 | <0.001 | 5.86 | <0.001 | 0.92 | NS |
| CAPB | | | 2.37 | NS | 0.17 | NS |
| Interaction terms | | | | | | |
| CAPB*2008–2010 | | | | | 0.96 | NS |
| CAPB*2012–2015 | | | | | 5.62 | <0.001 |
| Constant | 15.23 | <0.001 | 13.15 | <0.001 | 15.08 | <0.001 |
| **Preschool investment** | | | | | | |
| Independent variables | | | | | | |
| Time period | | | | | | |
| 2008–2010 | 0.1 | <0.001 | 0.1 | <0.001 | 0.1 | 0.006 |
| 2012–2013 | 0.18 | <0.001 | 0.18 | <0.001 | 0.18 | <0.001 |
| CAPB | | | –0.19 | NS | –0.19 | NS |
| Interaction terms | | | | | | |
| CAPB*2008–2010 | | | | | 0.008 | NS |
| CAPB*2012–2013 | | | | | –0.01 | NS |
| Constant | 0.72 | <0.001 | 0.85 | <0.001 | 0.85 | <0.001 |

SDCH (child poverty 2005–2015, material deprivation 2005–2015, and pre-school investment 2005–2013).
Reference category: years 2005–2007.
CAPB, cyclically adjusted primary balance; SDCH, social determinants of child health.

years immediately following the crisis, mainly in those countries with increasing needs in vulnerable population groups (ie, high unemployment rates). Afterwards, the Greek, Spanish and the UK governments modified public policy to reduce deficits by cutting public expenditure, particularly on welfare benefits, with adverse effects on SDCH. On the other hand, in other countries such as Iceland, besides its specific characteristics, governmental responses have attempted to give prominence to redistribution, and try to protect middle-income and low-income groups.[19] This fact would explain, in part, the differences observed between the austerity group to which Iceland belongs and SDCH indicators.

A systematic review of the impact of the crisis on population health included a few studies on child health,[13] one of which found increased odds of LBW deliveries during the crisis in Iceland.[26] It is worth noting that other variables should be taken into account when analysing perinatal outcomes such as LBW. For example, age of mother at birth, origin, social class, rates of multiple births, etc, although our data do not allow us to go further into these mechanisms affecting maternal environment and birth outcomes, which should be the subject of future studies. In any case, the results of our study are consistent with the finding of the mentioned study, particularly in the countries that applied the most severe austerity measures. The

studies included in the systematic review by Parmar *et al* are likely to have been too early to detect the full extent of the impact of the crisis and austerity measures on child health. Nevertheless, an increase in child mental health admissions and a high prevalence of mental health problems in children were described, and also poor well-being and an increase in the prevalence of overweight and obesity in poorest areas of the UK and among vulnerable children in Spain.[27–30]

A recent study showed that the rate of increase in life expectancy in 25 European countries slowed in 2011–2015 regarding 2006–2010.[31] The relationship between high austerity and LBW but not IM in the present study could be associated to lag effects, as suggested by a recent rise in IM in the UK over last 2 years,[32] and also the large numbers and much less power to detect changes in IM. These results reinforce the need for long-term monitoring of perinatal indicators such as IM.

This is one of the first studies to analyse trends in SDCH and perinatal indicators starting before the economic crisis. Although it is an observational study with ecological data from each country analysed, the findings provide credible evidence on the impact of austerity in childhood. The main strengths of our study include the use of nationally representative data with standardised and comparable definitions and methods, and a study design that allowed us to analyse time trends starting in 2005, 3 years before the economic crisis, through to 2015. Although the starting period, the extent and implementation of austerity measures might differ between countries and periods, the results are plausible and show a specific impact on SDCH and perinatal outcomes.

Some limitations of the study deserve comment. First of all, countries included in the study may not allow the results to be generalised. Nevertheless, European countries from the Euro and non-Euro areas were included. Future studies should extend the number of countries included to assess the influence of austerity in the remaining rich as well as middle-income and low-income countries. Second, those variables used to assess austerity, SDCH and perinatal outcomes may have limited ability to discriminate specific measures taken by governments. Moreover, validity and reliability of SDCH and perinatal indicators would be affected by country variability on data collection, etc. Nevertheless, both Eurostat and OECD make effort to continuously improve the quality of data collection (see http://ec.europa.eu/eurostat/en/web/products-statistical-working-papers/-/KS-RA-12-018 and https://data.oecd.org/healthstat/infant-mortality-rates.htm as examples). The austerity measure, the CAPB, used in this study to differentiate policy responses, although being a generally accepted economic measure of austerity, may not adequately reflect national policies which institute general economic stringency at the same time as attempting to protect the most vulnerable groups, such as children. This might explain why Iceland, although having a CAPB suggesting a high level of austerity, shows trends more consistent with those for countries with intermediate and low levels of austerity. Another limitation of these data is the absence of variables that allow inequalities in the SDCH and perinatal outcomes within countries to be analysed. It is well known that indicators analysed as averages often mask differences between population groups. Moreover, aggregate data do not allow adjusting LBW or IM by relevant variables such as parity, multiplicity or type of birth. In this sense, the sustained rise in multiple pregnancies associated with an increased access to assisted reproductive technology, a general deterioration in neonatal indicators such as LBW and a decrease in fertility since recession with changing socioeconomic status patterning, may underestimate the recession effect. For example, as previously mentioned, IM showed decreasing trends throughout the study period, but has been shown to have increased in the UK in recent years in the most disadvantaged social classes.[33] Third, the difficulties establishing causal associations between exposure and outcomes is well known especially in the case of international comparisons over time. Finally, as a result of the study design, the ecological fallacy cannot be ruled out.

In summary, to our knowledge, this is the first study to attempt to distinguish the negative impact on perinatal outcomes and SDCH of austerity measures imposed by government in response to the economic crisis from the effects of the economic crisis itself.

In conclusion, our findings suggest that those governments that applied high levels of austerity have exacerbated the effects of 2008 economic crisis on children specifically increasing child poverty, material deprivation in families at most need and perinatal outcomes such as LBW. The findings suggest the need to urgently protect vulnerable groups of children from the impact of austerity.

**Author affiliations**
[1]Retired, Barcelona, Spain
[2]Department of Public Health and Policy, University of Liverpool, Liverpool, UK
[3]Faculty of Social and Human Sciences, University of Iceland, Reykjavik, Iceland
[4]Centre for Health Equity Studies (CHESS), Stockholm University/Karolinska Institutet, Stockholm, Sweden
[5]Division of Mental Health and Wellbeing, Warwick Medical School, University of Warwick, Coventry, UK

**Contributors** LR and NS were involved in the initial conception and design of the study. LR extracted data and NS, DT-R, GG and AH participated in the data analysis strategy. All authors were involved in the interpretation and discussion of results through call conferences. LR developed the first draft of the manuscript and all authors critically revised it and approved the final version. LR is study guarantor.

**Funding** DT-R is funded by the MRC on a Clinician Scientist Fellowship (MR/P008577/1). The study was not externally financed.

**Competing interests** None declared.

**Patient consent** Not required.

**Ethics approval** Ethical approval was not required for this study as it used routinely available aggregated data.

**Provenance and peer review** Not commissioned; externally peer reviewed.

**Data sharing statement** The data are available on request to the corresponding author.

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
