## [Reviewer comments · BMJ Open]

ARTICLE DETAILS

TITLE (PROVISIONAL)	Trends in social determinants of child health and perinatal outcomes in European countries 2005-2015 by level of austerity imposed by governments: a repeat cross-sectional analysis of routinely available data
AUTHORS	Rajmil, Luis; Taylor-Robinson, David; Gunnlaugsson, Geir; Hjern, Anders; Spencer, Nick

VERSION 1 – REVIEW

REVIEWER	Johnathan Watkins PILAR Research Network, UK
REVIEW RETURNED	06-Apr-2018

GENERAL COMMENTS	Rajmil and colleagues have conducted a study to determine the temporal link between socioeconomic factors and perinatal outcomes in three time periods (2005-07, 2008-10 and 2012-15) and 18 countries. They concluded that countries that implemented more severe austerity measures have experienced increasing material deprivation among families with primary education as well as low birth weight, which has worsened the negative impact of economic crisis. As a consequence, they propose that to improve child health outcomes, austerity policies should be reversed. The authors are to be commended for their use of CAPB as a measure of austerity, since it adds a more robust basis for identifying which countries experienced austerity and which did not. However, the study requires some major questions answered and some clarifications. After these things have been done satisfactorily, this reviewer would consider re-reviewing the manuscript for publication. Major queries or comments (essential) 1. Why were these countries chosen, especially Canada and Australia? And why were other countries, especially from the OECD, not chosen? Preferably, there is an experimental design reason for this rather than a data constraint reason2. Are there any differences in the child poverty data between the EU-SILC and OECD Family Database for the European countries? The authors state that they took child poverty data for EU countries from EU-SILC and Aus and Can data from OECD Family Database. It would be good to know how comparable the data from these two differences. Simple way to do that is to do a statistical test for any differences between EU-SILC data and OECD Family Database data for those countries where both sources have data
--

	3. What happens to the results if only one time-based dummy variable is used? As the authors describe it, time is included in two dummy variables: one for year and a second for period (pre-crisis, crisis, and austerity or welfare). It's unclear to this reviewer whether this is a valid approach and whether it has any effect on the results 4. Clarify what the "2013-09" in "Table 1 shows the CAPB 2013-09" means 5. Provide labelling on Table 4 to show coefficients and p-values. Table 5 is labelled appropriately but Table 4 is not 6. Display Tables 2 and 3 visually instead of as Tables. It is not easy to follow what the authors are trying to describe at present Minor queries or comments (optional) 1. Change "Secondly" to "Second" and "Thirdly" to "Third" wherever used to be consistent with use of "First", which is grammatically correct 2. Provide the GEE model in equation form for clarity 3. Signpost the Results section with headings to show the three to five major takeaways 4. Provide the perinatal outcome trend as supplementary figures: a time series plot would show the trends well 5. Consider looking at whether using government spending on children yields the same results
--	--

REVIEWER	Carlos Varea PhD Associate Professor, Department of Biology, Faculty of Sciences, Autonomous University, Madrid (Spain)
REVIEW RETURNED	17-Apr-2018

GENERAL COMMENTS	The aim of the study—to evaluate the impact of the austerity policies associated with the economic recession on perinatal health—is of great interest, the objective of the paper is clear, and the analyses were adequately carried out using national, longitudinal aggregated data on perinatal outcome, i.e., including all babies born each year in each country (all parities, single/multiple gestations, types of birth, and maternal profile and origin). Prior to the recession, in almost all European countries there have been a sustained rise in multiple pregnancies associated with an increased access to Assisted reproductive technology, as well as an increased obstetric interventionism determining highest prevalence of Caesarean sections, trends explained as a consequence of the growing predominance of primipara mothers with an ever-increasing age at maternity. In this regard, to evaluate the negative impact of the economic recession of 2007/08 on perinatal health is challenging since during the preceding decade of sustained economic growth there was a general deterioration in neonatal indicators (except, mortality), e.g., LBW, in most European countries associated with the mentioned trends—as EURO-PERISTAT has argued. In contrast, the economic recession has determined a decrease of fertility in many countries, a decrease to which national women with a lower socio-economic status and immigrants are primarily contributing, with an increasing contribution to motherhood of highly qualified professionals and educated women, trends in the maternal profile than might underestimate the negative effects of the crisis on
--

	pregnancy and birth outcome. I agree with the conclusions of authors on evidences of a negative impact of the economic crisis (and, particularly, of austerity imposed by governments) on perinatal health, encouraging the acceptance of the paper, but I suggest that these general trends in maternal profile and obstetrics interventionism should be considered in the Introduction or the Discussion, as well as a mention in the Discussion on the limitations of aggregate data, which do not allow to adjust by relevant variables as parity, multiplicity or type of birth.
--	---

REVIEWER	Jane Sandall King's College, London, UK
REVIEW RETURNED	27-Apr-2018

GENERAL COMMENTS	This research aimed to assess whether the level of austerity implemented by national governments was associated with adverse trends in perinatal outcomes and the social determinants of children's health (SDCH) in rich countries. Longitudinal country level time trends in perinatal outcomes and SDCH from 2005-2015 (during three time periods 2005-7, 2008-10, and 2012-15) were studied in 16 European countries, plus Australia and Canada using available data from the International Monetary Fund (IMF), the Organisation for Economic Co-operation and Development (OECD), and Eurostat. Perinatal outcomes were low birth weight and infant mortality. Social determinants of child health were child poverty rates; severe material deprivation in families with primary education and pre-school investment. (GEE) models of repeated measures were used to assess time-trend differences in 3 periods. Countries with higher levels of austerity had worse outcomes, mainly at the last study period. The authors conclude that material deprivation increased during the period 2012-15 in those countries with higher CAPB, and that countries that implemented more severe austerity measures have experienced increasing material deprivation in families with primary education, and LBW, worsening the negative impact of economic crisis. Reversing austerity policies that impact children is likely to improve child health outcomes. This is an important paper. As the authors note this is one of few studies exploring the impact of austerity imposed by governments in high income countries on perinatal outcomes and SDCH above and beyond the impact of the economic crisis itself. As the authors point out, assessing impact of the financial crisis is challenging given the difficulty establishing association of exposure and health outcomes. There is a lag time between exposure and impact particularly with child outcomes and my comments are based on clarification of design, method and measurement. What was rationale for choice of countries and why exclusion of USA? Why these perinatal outcomes? Was this what was available or is
---

	this testing hypothesises from previous work? Clarify whether LBW was less than but not including 2500gm? SDCH are not really outcomes but process indicators. What was the rationale to use these rather than child health outcomes? What data quality assessments were conducted? Can authors comment on why there was a relationship between high austerity on some outcomes such as LBW but not others such as IM? Major limitation is ability to differentiate between economic austerity and the policy responses. The austerity measure (the CAPB) used in this study, although being a generally accepted economic measure of austerity, may not adequately reflect national policies which institute general economic stringency. The findings on Iceland in the discussion would benefit from more attention in the results to illustrate that policies to moderate austerity seem to have some success.
--	--

REVIEWER	Dr. Musa Abubakar Kana Department of Epidemiology and Community Medicine, Federal University Lafia, Nasarawa State, Nigeria
REVIEW RETURNED	16-Jun-2018

GENERAL COMMENTS	The authors addressed the important topic of the association of austerity imposed by governments after the economic crisis with social determinants of child health and perinatal outcomes in rich countries. The authors investigated the hypotheses that austerity was negatively associated with social determinants of child health and perinatal outcomes. The authors used secondary data to perform longitudinal analysis by making comparisons according to level of austerity imposed by governments relative to period before economic crisis (2005-07), economic crisis (2008-12) and during austerity (2012-15). They observed a negative association between austerity and social determinants of child health in countries with high austerity. But there is no distinctive effect of the austerity measures on low birthweight and infant mortality. Although, the methodology the authors employed demonstrated these associations but it was not a conclusive causal association because there are important limitations. It is an ecological study design that used secondary data. The analysis considered only exposure variable (austerity measure) and outcome variables (social determinant of child health and prenatal outcomes). Only one covariate (time) was considered in the generalizing estimating equation. In the absence of other important variables with confounding or effect modifying properties. It cannot be sufficiently concluded that austerity on its own mediated the negative changes in social determinants of child health and prenatal outcomes. The manuscript can be improved if the authors undertake revision of the manuscript to address the following issues. Abstract Introduction  • The perinatal indicators (low birth weight and infant mortality) considered in this study have multiple determinants (individual and contextual level). It is important that the authors highlight the
---

pathway of the association between austerity measures (time context) and expected changes in these outcomes. What is the role of individual mediating and other contextual factors?

- Social determinant of child health is a multidimensional concept. It is vital that the authors unbundle these determinants and specify those that are sensitive (and how) to austerity measures.
- Both economic crisis and austerity measures have effect on population health. The authors need to differentiate how the two economic phenomena influence population health. Does austerity act independently? or it is an effect modifier to the influence of economic crisis on prenatal outcomes or social determinants of child health?
- The authors need to reorganize the introduction to clearly articulate the issues identified above in logically supportive paragraphs.

Methodology

- Analysis
 - o The time trend categories used in the study as illustrated in Tables 2 and 3 were 2005-07, 2008-10 and 2012-15. How about 2011?
 - o Can the authors include a table comparing important individual level and contextual factors (GDP and social spending) for the selected countries in the 3 time trend categories?
 - o There are many published time trend studies employing different statistical approaches to investigate the effect of 2008 economic crisis on prenatal indicators. The authors need to elaborate on their justification for the method they used in this study. Can other covariates be adjusted in the prediction model?

Discussion

- In the second paragraph of the discussion the authors asserted that, “the initial response to the crisis was increasing public spending during the years immediately following the crisis”. But there are no results in this study to support the statement. It is recommended in the methodology section that the authors should include a table with data on GDP and social spending
- Is the austerity directly associated with perinatal outcomes and social determinants of child or the association is mediated by other variables?
- The authors need to distinguish whether or not economic crisis has an initial impact on prenatal outcomes and sustained after implementation of austerity.
- The authors need to discuss more on why there is reduction of infant mortality during the economic crisis and austerity
- The following need to be added to the study limitations
 - o Was the time period (2 years) before the economic crisis sufficient to provide evidence about the situation?
 - o The relevance of ecological fallacy in the generalization of study findings
 - o The package of austerity, beginning and extend of implementation might differ between countries, which could have effect on the measurement and classification of the exposure variable.

Conclusion

- There is no clearly written conclusion. The authors need to specify their major finding as it relates to their aim and hypothesis.

VERSION 1 – AUTHOR RESPONSE

Reviewers' Comments to Author:

Reviewer: 1

Reviewer Name: Johnathan Watkins

Institution and Country: PILAR Research Network, UK Competing Interests: None declared

Rajmil and colleagues have conducted a study to determine the temporal link between socioeconomic factors and perinatal outcomes in three time periods (2005-07, 2008-10 and 2012-15) and 18 countries. They concluded that countries that implemented more severe austerity measures have experienced increasing material deprivation among families with primary education as well as low birth weight, which has worsened the negative impact of economic crisis. As a consequence, they propose that to improve child health outcomes, austerity policies should be reversed.

The authors are to be commended for their use of CAPB as a measure of austerity, since it adds a more robust basis for identifying which countries experienced austerity and which did not. However, the study requires some major questions answered and some clarifications. After these things have been done satisfactorily, this reviewer would consider re-reviewing the manuscript for publication.

Re: Authors would like to thank the Reviewer for this comment.

Major queries or comments (essential)

1. Why were these countries chosen, especially Canada and Australia? And why were other countries, especially from the OECD, not chosen? Preferably, there is an experimental design reason for this rather than a data constraint reason

Re: We would like to thank the Reviewer for this specific comment. As the reviewer point out it was not a data constrain but an experimental design reason for countries included in the study. As commented in the introduction section of the study, assessing the health effects of the crisis is challenging given the difficulty establishing association of exposure and outcomes. Moreover, beside factors analysed in the present study, some known and unknown factors may have influenced exposure and government responses to the crisis at country level. In order to try to minimize other potential confounders the proposal included European countries from Euro and non-Euro area. Then, countries such as Luxembourg were excluded due to its high level of economic development, which not necessarily reflects the real wealth of residents. Post-communist countries were also excluded. Australia and Canada were excluded in the revised version following the Editor and Reviewers' comments in order to be more consistent and to centre our analysis on European countries, and the data have been re-analysed. According to this comment and also a comment from the Reviewer 3 and an Editorial request, a paragraph has been added on the inclusion criteria to the method section (see page 5, first paragraph in the revised version and also the answers to Editorial requests):

*"Sixteen European countries were included in the study, using routinely available data for the period 2005-2015 pertaining to perinatal outcomes and SDCH."*². Are there any differences in the child poverty data between the EU-SILC and OECD Family Database for the European countries? The authors state that they took child poverty data for EU countries from EU-SILC and Aus and Can data from OECD Family Database. It would be good to know how comparable the data from these two differences. Simple way to do that is to do a statistical test for any differences between EU-SILC data and OECD Family Database data for those countries where both sources have data

Re: Following previous comments Australia and Canada have been excluded in the revised version of the manuscript. As a consequence, there is only one data source for each indicator.

3. What happens to the results if only one time-based dummy variable is used? As the authors describe it, time is included in two dummy variables: one for year and a second for period (pre-crisis, crisis, and austerity or welfare). It is unclear to this reviewer whether this is a valid approach and whether it has any effect on the results

Re: We apologise for the mistake in the explanation of the Generalised Estimated Equation (GEE) modelling. We included time period in the final model in 3 time-period groups, and this has now been clarified in the methods (page 6 - Data analysis):

“Finally we used a longitudinal Generalised Estimating Equations (GEE) model, with model-based on robust standard errors²³, to allow for analysis of our correlated repeated outcome measurements (SDCH and perinatal outcomes). The independent variables included in the models were time-period, and CAPB. Time-period was included in the model as a yearly dummy variable (2005 to the latest available data), and stratified in 3 time periods. Interaction terms were also included in the model to assess the influence of the time-period and the level of austerity. The analysis was conducted using STATA version 12.0.

4. Clarify what the 2013-09 in Table 1 shows the CAPB 2013-09 means

Re: A footnote clarifying the meaning of the difference 2013-09 in the Cyclically Adjusted Primary Balance (CAPB) has been added (see Table 1 in the revised version)

5. Provide labelling on Table 4 to show coefficients and p-values. Table 5 is labelled appropriately but Table 4 is not

Re: Labelling on Table 4 has been added (see Table 4 in the revised version)

6. Display Tables 2 and 3 visually instead of as Tables. It is not easy to follow what the authors are trying to describe at present

Re: Thank you - we have now added figures in the supplementary material including yearly SDCH and perinatal outcomes at individual country level (see Figures 2a-b and 3a-c of the Supplementary material in the revised version) to supplement the data in the tables.

Minor queries or comments (optional)

1. Change Secondly to Second and Thirdly to Third wherever used to be consistent with use of First, which is grammatically correct

Re: As recommended by this Reviewer and also following an Editorial request, a complete revision of the language edition has been made.

2. Provide the GEE model in equation form for clarity

Re: Thank you – we have added a description of the model in the supplementary material:

Figure 1 in the Supplementary material:

GEE approach is an extension of Generalising linear models that provides a semi-parametric approach to longitudinal analysis of categorical and continuous measurements. In this study, a GEE model with link identity and normal distribution, and exchangeable working correlation structure was selected. Given the response for country i at time-period j , and the $p \times 1$ vector of covariates, the marginal response is estimated as:

$$E = \beta$$

Where β is a $p \times 1$ vector of regression coefficients. The (j,k) element of the working correlation matrix R_i is the estimated correlation between and

$$Corr =$$

3. Signpost the Results section with headings to show the three to five major takeaways

Re: Thanks - the revised results section includes 2 headings to show the main results of the study (please see page 7 in the Results section):

Perinatal and SDCH outcomes according to the level of austerity

Multivariate models

4. Provide the perinatal outcome trend as supplementary figures: a time series plot would show the trends well

Re: Thanks – we have included a figure of trends in perinatal outcomes and SDCH by country and year as Supplementary material (see Figures 2a-b and 3a-c on the Supplementary material in the revised version).

5. Consider looking at whether using government spending on children yields the same results

Re: Thank you – we have already used pre-school investment in children as one of the outcomes in our GEE model to assess how Cyclically Adjusted Primary Balance, our measure of recession, is associated with government spending on children.

Reviewer: 2

Reviewer Name: Carlos Varea PhD

Institution and Country: Associate Professor, Department of Biology, Faculty of Sciences, Autonomous University, Madrid (Spain) Competing Interests: None declared

The aim of the study to evaluate the impact of the austerity policies associated with the economic recession on perinatal health is of great interest, the objective of the paper is clear, and the analyses were adequately carried out using national, longitudinal aggregated data on perinatal outcome, i.e., including all babies born each year in each country (all parities, single/multiple gestations, types of birth, and maternal profile and origin).

Prior to the recession, in almost all European countries there have been a sustained rise in multiple pregnancies associated with an increased access to Assisted reproductive technology, as well as an increased obstetric interventionism determining highest prevalence of Caesarean sections, trends explained as a consequence of the growing predominance of primipara mothers with an ever-increasing age at maternity. In this regard, to evaluate the negative impact of the economic recession of 2007/08 on perinatal health is challenging since during the preceding decade of sustained economic growth there

was a general deterioration in neonatal indicators (except, mortality), e.g., LBW, in most European countries associated with the mentioned trends as EURO-PERISTAT has argued. In contrast, the economic recession has determined a decrease of fertility in many countries, a decrease to which national women with a lower socio-economic status and immigrants are primarily contributing, with an increasing contribution to motherhood of highly qualified professionals and educated women, trends in the maternal profile than might underestimate the negative effects of the crisis on pregnancy and birth outcome.

Re: Authors would like to thank the Reviewer for this important comment. It is true that specific perinatal outcomes such as LBW showed a trend to increase during the period previous to the economic crisis in several European countries. Also, some factors such as age of mother at birth, origin, social class, rates of multiple births, etc. should be taken into account looking at trends on this indicator. Unfortunately this information is not available on International aggregated databases. Nevertheless, and even in the case that at the end of the period the contribution of these factors would improve LBW, this fact would be in the opposite direction than the study hypothesis: presumably those countries with high impact of the economic crisis and high austerity government responses would improve LBW more than the rest of countries. And the results showed increasing rates of LBW in countries with high austerity level. Of course, the ecological design of the study precludes being absolutely sure on these conclusions. According to this comment a sentence has been added to the discussion section on this issue (see page 8, paragraph 3 in the Discussion section):

“A systematic review of the impact of the crisis on population health included a few studies on child health,¹³ one of which found increased odds of LBW deliveries during the crisis in Iceland.²⁵ the results of our study are consistent with this finding , particularly”

I agree with the conclusions of authors on evidences of a negative impact of the economic crisis (and, particularly, of austerity imposed by governments) on perinatal health, encouraging the acceptance of the paper, but I suggest that these general trends in maternal profile and obstetrics interventionism should be considered in the Introduction or the Discussion, as well as a mention in the Discussion on the limitations of aggregate data, which do not allow to adjust by relevant variables as parity, multiplicity or type of birth.

Re: A specific sentence on this issue has been added to the limitations of the study (see page 9 paragraph 2 in the Discussion section):

“...Another limitation of this data is the absence of variables that allow inequalities in the SDCH and perinatal outcomes within countries to be analysed. It is well known that indicators analysed as averages often mask differences between population groups”.

Reviewer: 3

Reviewer Name: Jane Sandall

Institution and Country: King's College, London, UK Competing Interests: None declared

This research aimed to assess whether the level of austerity implemented by national governments was associated with adverse trends in perinatal outcomes and the social determinants of children s health (SDCH) in rich countries. Longitudinal country level time trends in perinatal outcomes and SDCH from 2005-2015 (during three time periods 2005-7, 2008-10, and 2012-15) were studied in 16 European countries, plus Australia and Canada using available data from the International Monetary Fund (IMF), the Organisation for Economic Co-operation and Development (OECD), and Eurostat.

Perinatal outcomes were low birth weight and infant mortality. Social determinants of child health were child poverty rates; severe material deprivation in families with primary education and pre-school investment. (GEE) models of repeated measures were used to assess time-trend differences in 3 periods. Countries with higher levels of austerity had worse outcomes, mainly at the last study period. The authors

conclude that material deprivation increased during the period 2012-15 in those countries with higher CAPB, and that countries that implemented more severe austerity measures have experienced increasing material deprivation in families with primary education, and LBW, worsening the negative impact of economic crisis. Reversing austerity policies that impact children is likely to improve child health outcomes.

This is an important paper. As the authors note this is one of few studies exploring the impact of austerity imposed by governments in high income countries on perinatal outcomes and SDCH above and beyond the impact of the economic crisis itself. As the authors point out, assessing impact of the financial crisis is challenging given the difficulty establishing association of exposure and health outcomes. There is a lag time between exposure and impact particularly with child outcomes and my comments are based on clarification of design, method and measurement.

What was rationale for choice of countries and why exclusion of USA?

Re: We would like to thank the Reviewer for this comment. As commented in the introduction section of the study, assessing the health effects of the crisis is challenging given the difficulty establishing association of exposure and outcomes. Moreover, beside factors analysed in the present study, some known and unknown factors may have influenced exposure and government responses to the crisis at country level. In order to try to minimize other potential confounders the initial proposal included European countries from Euro and non-Euro area. Then countries such as Luxembourg were excluded due to its high level of economic development, which not necessarily reflects the real wealth of residents. Post-communist countries were also excluded. Australia and Canada were excluded in the revised version following the Editor and Reviewers' comments in order to be more consistent and to center our analysis into European countries, and the data have been re-analysed. According to this comment and also a comment from the Reviewer 1 and an Editorial request, a paragraph has been added on the inclusion criteria to the method section (see page 5, first paragraph in the revised version and also the answers to Editorial requests):

"Sixteen European countries were included in the study routinely available data for the period 2005-2015 pertaining to perinatal outcomes and SDCH.

Why these perinatal outcomes? Was this what was available or is this testing hypotheses from previous work?

Re: According to the literature review and our previous work on this subject foetal and childhood population are particularly vulnerable population groups in the context of economic crisis. Following to the initial review and also a systematic review carried out by our group (see Ref 12 in the revised version) we focused on important perinatal outcomes for which data was available at aggregate level in our analysis. We chose a mortality indicator (IM) and an outcome that is associated with subsequent life chances (LBW). This information has been included in the introduction of the manuscript (see page 4 paragraph 3 in the introduction section):

"Findings from studies to date on the impact of the economic crisis on perinatal outcomes and social determinants of child health (SDCH), carried out at national and international level show foetal and childhood populations as population groups most affected by the crisis."¹²⁻¹⁶

Clarify whether LBW was less than but not including 2500gm?

SDCH are not really outcomes but process indicators. What was the rationale to use these rather than child health outcomes?

What data quality assessments were conducted?

Re: LBW was defined as less than 2500gr according to the OECD Family Database (see page 5 perinatal outcomes subheading in the Methods section).

SDCH were included according to a model initially developed by Dahlgren and Whitehead (see Ref 8 in the revised manuscript) where these determinants are associated to the outcomes.

Moreover, it is known that the exposure to high level of poverty, economic stress and /or higher level of inequalities at early ages are associated to worse health for the future adult population.

We have included SDCH based on these time lags from changes in SDCH to outcomes, and longer term outcomes. Regarding quality assessment of the available data, both Eurostat and OECD carry out periodic assessments trying to improve data collection, sampling design, etc.

(see for example: <http://ec.europa.eu/eurostat/en/web/products-statistical-working-papers/-/KS-RA-12-018> and <https://data.oecd.org/healthstat/infant-mortality-rates.htm>). According to this comment a sentence has been added to the limitations of the study (see page 9 paragraph 2):

“Secondly, those variables used to assess austerity, SDCH, and perinatal outcomes may have limited ability to discriminate specific measures taken by Governments.

Can authors comment on why there was a relationship between high austerity on some outcomes such as LBW but not others such as IM?

Re: Initially we expected to find an important negative impact of the crisis and the level of austerity on SDCH but less impact on perinatal outcomes. In fact IM showed a decreasing trend during the study period in almost all countries. Nevertheless, in this study we analysed IM as average at country level which is likely to mask differences between population groups as happened in Greece or UK. The relationship between high austerity and LBW but not IM could be associated to lag effects, as suggested by recent rise in IM in the UK over the last 2 years , and also the large numbers and much less power to detect changes in IM. A sentence has been added to the discussion section (see page 8, last paragraph):

“ A systematic review... in Spain. ²⁶⁻²⁹

Major limitation is ability to differentiate between economic austerity and the policy responses. The austerity measure (the CAPB) used in this study, although being a generally accepted economic measure of austerity, may not adequately reflect national policies which institute general economic stringency. The findings on Iceland in the discussion would benefit from more attention in the results to illustrate that policies to moderate austerity seem to have some success.

Re: We would like to thank the Reviewer for this comment. It is true that it is difficult to differentiate between the impact of the economic crisis and the policy responses (see Ref 6 in the revised version). And also CAPB may not reflect some specific policy responses. As the reviewer point out, the case of Iceland may illustrate the implementation of austerity with different results, although we think that the case of Iceland is quite specific, as it was commented through the manuscript. According to this comment a sentence has been modified (see page 9 paragraph 2 in the Discussion section):

“The austerity measure, the CAPB, used in this study, although being a generally accepted economic measure of austerity, may not reflect national policies which institute general economic stringency at the same time as attempting to protect the most vulnerable groups, such as children. This might ~~partly~~ explain why Iceland, although having a CAPB suggesting a high level of austerity, shows trends more consistent with those for countries with intermediate and low levels of austerity.”

Reviewer: 4

Reviewer Name: Dr. Musa Abubakar Kana

Institution and Country: Department of Epidemiology and Community Medicine, Federal University Lafia, Nasarawa State, Nigeria
Competing Interests: None declared

The authors addressed the important topic of the association of austerity imposed by governments after the economic crisis with social determinants of child health and perinatal outcomes in rich countries. The authors investigated the hypotheses that austerity was negatively associated with social determinants of child health and perinatal outcomes. The authors used secondary data to perform longitudinal analysis by making comparisons according to level of austerity imposed by governments relative to period before economic crisis (2005-07), economic crisis (2008-12) and during austerity (2012-15). They observed a negative association between austerity and social determinants of child health in countries with high austerity. But there is no distinctive effect of the austerity measures on low birthweight and infant mortality. Although, the methodology the authors employed demonstrated these associations but it was not a conclusive causal association because there are important limitations. It is an ecological study design that used secondary data. The analysis considered only exposure variable (austerity measure) and outcome variables (social determinant of child health and prenatal outcomes). Only one covariate (time) was considered in the generalizing estimating equation. In the absence of other important variables with confounding or effect modifying properties. It cannot be sufficiently concluded that austerity on its own mediated the negative changes in social determinants of child health and prenatal outcomes. The manuscript can be improved if the authors undertake revision of the manuscript to address the following issues.

Abstract**Introduction**

The perinatal indicators (low birth weight and infant mortality) considered in this study have multiple determinants (individual and contextual level). It is important that the authors highlight the pathway of the association between austerity measures (time context) and expected changes in these outcomes. What is the role of individual mediating and other contextual factors? Social determinant of child health is a multidimensional concept. It is vital that the authors unbundle these determinants and specify those that are sensitive (and how) to austerity measures. The authors need to reorganize the introduction to clearly articulate the issues identified above in logically supportive paragraphs.

Re: We would like to thank the Reviewer for this comment. It is true that perinatal outcomes such as infant mortality and LBW have multiple determinants, at individual and contextual level. Our study was focused on contextual factors following the model developed by Dahlgren and Whitehead (see Ref 8 in the revised manuscript). These contextual factors such as living conditions, family income, employment, education, housing, access to health services, etc., are the social determinants of child health, and have an impact on health. Published studies on the economic crisis showed that foetal and childhood population were one of the population groups most affected by the crisis. Moreover, children are dependent on family factors including financial capital (family income), human capital (education) and social capital (family working conditions) (see Ref 7 in the revised version of the manuscript). Our hypothesis was that, once the economic crisis was overcome at macroeconomic level, those countries that implemented and maintained more austerity measures after the economic crisis (cutting budget in social welfare, and in unemployment, implementing barriers to access to healthcare services, etc.) showed worse social determinants of child health and consequently worse perinatal outcomes than countries at lower level of austerity. We have tried to reorganise the theoretical model in which it was based our work in the introduction section as follow (see page 3 paragraphs 2 and 3):

“Assessing the health effects of the crisis is challenging given the difficulty establishing association of exposure and outcomes. There is a lag time between exposure to the effects of a severe economic downturn and adverse health outcomes⁵ and, in the case of exposure in early childhood, the adverse effects may not emerge until adulthood.⁶ Childhood is an especially vulnerable period to the main determinants of health, ~~from socioeconomic and environmental macro conditions, and have an impact on health,~~ to individual life style factors. All factors related to financial capital (i.e. family income), human capital (i.e. education), and social capital (family working conditions) have potential influence on future child health and development.^{7,8} Moreover, there is overwhelming evidence for the profound effects of social factors on health throughout childhood and into adulthood.⁹⁻¹¹

Findings from studies to date on the impact of the economic crisis on perinatal outcomes ~~and social determinants of child health (SDCH)~~, carried out at national and international level show foetal and childhood populations as population groups most affected by the crisis.¹²⁻¹⁶ Vulnerability starts in the prenatal period with adverse effects on perinatal outcomes.¹⁷⁻¹⁹ Most studies of the impact of economic crisis on health have not distinguished between economic crises themselves and policy responses to these crises.⁶ Furthermore there is a paucity of literature on the impact of austerity measures imposed by government in response to the economic crisis on perinatal outcomes and SDCH.²⁰ .”

Both economic crisis and austerity measures have effect on population health. The authors need to differentiate how the two economic phenomena influence population health. Does austerity act independently? or it is an effect modifier to the influence of economic crisis on prenatal outcomes or social determinants of child health?

Re: We agree with the Reviewer on the need and difficulty of distinguishing between crisis and policy responses. In this sense we have included the measure of austerity (the CAPB) both as independent variable and as effect modifier (see models 2 and 3 in Tables 4 and 5). As our hypothesis was to check specific time-periods the main interest was to analyse the interaction effect of austerity with the last study period (2012-2015) compared to previous periods. This is why the main results showed were (see the abstract): *Material deprivation increased during the period 2012-15 in those countries with higher CAPB (interaction CAPB-period 2012-15, B: 5.62; p< 0.001), as did LBW (interaction CAPB-period 2012-15, B: 0.25; p 0.004).* We think that these results give plausibility to the study hypothesis. Introduction to the interaction terms was included in the statistical analysis (see the Data analysis, page 6, paragraph 5, last sentence in the revised version):

“...Interaction terms were also included in the model to assess the influence of the time-period and the level of austerity...”

Methodology

Analysis

o The time trend categories used in the study as illustrated in Tables 2 and 3 were 2005-07, 2008-10 and 2012-15. How about 2011?

Re: Time trend categories were approached trying to disentangle as much as possible those periods with higher impact of the crisis and government responses. In this sense, and although there were high variability between countries regarding time periods it is universally accepted that the main impact of the crisis was the years 2008 to 2010. We have considered that starting the last period in 2012 would better reflect the level of austerity implemented by governments after the crisis.

o Can the authors include a table comparing important individual level and contextual factors (GDP and social spending) for the selected countries in the 3 time trend categories?

Re: We would like to thank the Reviewer for this comment. As contextual variable it was included pre-school investment in children 0-5y that was considered one of the main variables with impact on childhood living conditions (see table 3). Other variables such as investment on unemployment, etc. were not included given their relative high correlation to the austerity measure used (CAPB).

o There are many published time trend studies employing different statistical approaches to investigate the effect of 2008 economic crisis on prenatal indicators. The authors need to elaborate on their justification for the method they used in this study. Can other covariates be adjusted in the prediction model?

Re: We would like to thank the Reviewer for this comment. We think that the Generalised Estimating Equation modelling represents a good approach to check our hypotheses, and have added a citation to the use of GEE models for evaluation policy changes/natural experiments (see ref 24 in the revised version). Unfortunately the lack of specific variables such as age of mothers, multiplicity, etc., does not allow adjusting by other relevant variables. According to this comment and also a comment from the Reviewer 2 a sentence has been added to the limitations of the study (see page 9 paragraph 2 in the revised version)

“...Another limitation of this data is the absence of variables that allow inequalities in the SDCH and perinatal outcomes within countries to be analysed. It is well known that indicators analysed as averages often mask differences between population groups. ”.

Discussion

In the second paragraph of the discussion the authors asserted that, the initial response to the crisis was increasing public spending during the years immediately following the crisis . But there are no results in this study to support the statement. It is recommended in the methodology section that the authors should include a table with data on GDP and social spending

Re: We agree with the Reviewer that increasing public spending was not a measure given in all countries included in the study at the beginning of the crisis. It depends on the impact on vulnerable population and the level of social protection provided.

On the other hand, we have checked GDP per capita 2005-2015 yearly (see the Table below) and its influence on the results. As it can be checked there was not so high variability of the GDP within and between countries in the study period

GDP per capita in US Dollars (Source: OECD database)

Year/country	2005	2006	2007	2008	2009	2010	2011	2012	2013	2014	2015
Austria	35 025	37 631	39 387	41 316	40 928	42 059	44 469	46 478	47 937	48 801	49 959
Belgium	33 331	35 391	36 865	38 133	38 004	40 089	41 450	42 585	43 746	44 717	45 561
Denmark	34 153	37 302	38 972	41 283	40 333	43 047	44 408	44 809	46 743	47 905	48 688
Finland	31 993	34 367	37 697	39 969	37 823	38 775	40 683	40 620	41 293	41 463	42 064
France	30 504	32 445	34 090	35 103	34 687	35 944	37 448	37 684	39 529	40 145	40 565
Germany	32	34	37	38	37	39	42	43	45	47	47

	414	754	018	663	689	955	693	564	232	092	811
Greece	25 577	28 523	29 287	30 856	30 360	28 176	26 141	25 284	26 098	26 839	26 697
Iceland	37 069	38 652	40 802	42 456	41 105	38 540	39 624	40 698	42 817	44 544	47 502
Ireland	40 437	44 250	46 738	44 217	41 579	43 302	45 176	46 469	48 297	51 468	69 459
Italy	29 938	32 179	33 789	35 155	34 228	34 719	35 935	35 757	35 885	36 071	36 640
Netherlands	37 272	40 615	43 481	45 859	44 049	44 552	46 067	46 716	48 679	48 612	49 551
Norway	47 775	54 083	55 889	61 760	55 424	58 025	62 146	65 442	67 051	66 018	61 713
Portugal	22 740	24 659	25 701	26 631	26 464	27 335	26 780	26 454	27 899	28 747	29 532
Spain	27 696	30 845	32 585	33 443	32 382	31 964	32 073	31 993	32 623	33 728	34 844
Sweden	33 967	37 423	40 573	41 854	39 646	41 628	43 755	44 725	45 673	46 524	47 891
United Kingdom	32 431	34 512	35 291	36 248	34 599	35 880	36 593	37 703	39 322	40 717	41 592

We have also analysed GDP in the 3 time periods analysed and we have found no changes in the models when introducing baseline GDP 2005-2007 as an independent variable in the GEE models, probably because this low level of variability and also given that the independent variable (CAPB) explain an important part of the adjusted model on economic changes. As an example, find below a comparison of the final model of material deprivation with and without adjustment by GDP 2005-2007 (Table 5 of the revised version):

Table 5

Material deprivation	Model 3 Interactions (time-period*CAPB)	
	Coef.	P
Independent variables		
Time period		
2008-10	0.01	NS
2012-15	0.92	NS
CAPB	0.17	NS
Interaction terms		
CAPB*2008-10	0.96	NS
CAPB*2012-15	5.62	<0.001
Constant	15.08	<0.001

Table 5 adjusted by GDP at baseline (2005-2007)

Material deprivation	Model 3 Interactions (time-period*CAPB)	
	Coef.	p
Independent variables		
Time period		
2008-10	0.01	NS
2012-15	0.92	NS
CAPB	-0.7	NS
GDP 2005-2007	-0.003	NS

Interaction terms		
CAPB*2008-10	0.96	NS
CAPB*2012-15	5.62	<0.001
Constant	28.0	<0.001

Also, similar results were found with other SDCH and perinatal outcomes. As a consequence we preferred to avoid including GDP in the models and also as a Table. According to this comment from the Reviewer the second paragraph in the discussion section has been modified as follow (see page 8 paragraph 2 in the discussion section):

“...In general, the initial response to the crisis was increasing public spending during the years immediately following the crisis, .”

Is the austerity directly associated with perinatal outcomes and social determinants of child or the association is mediated by other variables?

Re: As it was stated previously variables such as living conditions, family income, employment, education, housing, and access to health services, among other social determinants of child health, have an impact on child health. Following the Dahlgren and Whitehead model these factors may influence individual behaviours and also impact on child health. We have added a sentence to the manuscript to reflect this in the introduction section (page 4 paragraph 3):

“...Most studies of the impact of economic crisis on health have not distinguished between economic crises themselves and policy responses to these crises.⁶ Furthermore there is a paucity of literature on the impact of austerity measures imposed by government in response to the economic crisis on perinatal outcomes and SDCH.²⁰

The authors need to distinguish whether or not economic crisis has an initial impact on prenatal outcomes and sustained after implementation of austerity.

Re: As it was stated in the introduction section the initial literature review showed an impact of the crisis on specific perinatal outcomes such as low birth weight in some countries (see ref 12 to 19 in the revised version).

The authors need to discuss more on why there is reduction of infant mortality during the economic crisis and austerity

Re: Infant mortality rates in high-income countries in the last decades shows a trend to progressively diminish. This fact is universally accepted as related to the welfare state, increase prenatal and postnatal access to healthcare services, and other social factors. In our study IM showed a slightly decrease during the study period in almost all countries. Nevertheless we analysed IM as average at country level and this fact often mask differences between population groups as happened in Greece or UK. Moreover, it would be possible that IM will be affected in the long term and especially if the reduction of family and social benefits and barriers to access to perinatal care continues in some countries. A recent published study on life expectancy in 25 European countries showed that life expectancy increase slowed in almost all the countries studied in the period 2011-2015 regarding 2006-2010, but the slowdown was particularly marked in specific countries such as the UK (Marmot M. Social causes of the slowdown in health improvement. J Epidemiol Community Health. 2018; doi: 10.1136/jech-2018-210580). A sentence has been added to the Discussion section on these issues (see page 8, last paragraph in the revised version):

“...Nevertheless, an increase in child mental health admissions and a high prevalence of mental health problems in children were described, and also poor well-being and an increase in the prevalence of overweight and obesity in poorest areas of the UK and among vulnerable children in Spain.”²⁶⁻²⁹

The following need to be added to the study limitations

- o Was the time period (2 years) before the economic crisis sufficient to provide evidence about the situation?

Re: We think that study periods are comparable. The study periods analysed were 2005 to 2007 (3 year) before the economic crisis; the economic crisis (2008 to 2010, 3 years) and the austerity period 2012 to 2015 (4 years): We think that all 3 study periods are balanced and easy to be compared.

- o The relevance of ecological fallacy in the generalization of study findings

Re: We would like to thank the Reviewer for this comment. A sentence has been added to the limitations of the study (see page 9, second to the last paragraph in the Discussion section):

“Thirdly, the difficulties establishing causal associations between exposure and outcomes is well known especially in the case of international comparisons over time.”

- o The package of austerity, beginning and extend of implementation might differ between countries, which could have effect on the measurement and classification of the exposure variable.

Re: A sentence has been added in the Discussion section regarding this issue (see page 9, first paragraph in the Discussion section):

“... The main strengths of our study include the use of nationally representative data with standardised and comparable definitions and methods, and a study design that allowed us to analyse time trends starting in 2005, 3 years before the economic crisis, through to 2015.”

Conclusion

There is no clearly written conclusion. The authors need to specify their major finding as it relates to their aim and hypothesis.

Re: The conclusion has been modified as follows (see page 10, first paragraph):

“In summary..... crisis itself.

Our findings suggest that those governments that applied high levels of austerity have exacerbated the effects of 2008 economic crisis specifically increasing child poverty, material deprivation in families at most need, and perinatal outcomes such as LBW. The findings suggest the need to urgently protect vulnerable groups of children from the impact of austerity.”

FORMATTING AMENDMENTS (if any)

Required amendments will be listed here; please include these changes in your revised version:

-Authors must include a statement in the Methods section of the manuscript under the sub-heading 'Patient and Public Involvement'.

This should provide a brief response to the following questions:

- How was the development of the research question and outcome measures informed by patients priorities, experience, and preferences?
- How did you involve patients in the design of this study?
- Were patients involved in the recruitment to and conduct of the study?
- How will the results be disseminated to study participants?
- For randomised controlled trials, was the burden of the intervention assessed by patients themselves?
- Patient advisers should also be thanked in the contributorship statement/acknowledgements.

If patients and or public were not involved please state this.

Re: A subheading has been added to the Methods section on “Patient and Public Involvement” (please see page 6, last paragraph in the revised version):

VERSION 2 – REVIEW

REVIEWER	Johnathan Watkins PILAR Research Network, UK
REVIEW RETURNED	23-Jul-2018

GENERAL COMMENTS	The authors have responded satisfactorily to my comments on their first submission. The paper is now much improved.
---

REVIEWER	Varea, Carlos Department of Biology, Faculty of Sciences, Madrid Autonomous University, Madrid, Spain
REVIEW RETURNED	26-Jul-2018

GENERAL COMMENTS	The new version of the manuscript by Rajmil et al. has improved in quality and clarity. Congratulations. Authors have incorporated considerations on trends in birth outcome previous to the economic crisis as well as some limitations of aggregate data suggested in my previous review. In the revised version, 2012-15 is defined as “post-crisis period”. At least for Spain, it is far from obvious. Perhaps it is more appropriate to describe these years as of “limited economic recovery” compared with the worst years of recession. A further study on the impact of the economic crisis on LBW in Spain according to the socioeconomic status of the mothers is already available: Terán et al. 2018. Social disparities in Low Birth Weight among Spanish mothers during the economic crisis (2007-2015). Nutr Hosp 35 (N.º Extra. 5):129-141. Thanks
--